

# Determining insulin sensitivity from glucose tolerance tests in Iberian and landrace pigs

José Miguel Rodríguez-López[1], Manuel Lachica[2], Lucrecia González-Valero[2] and Ignacio Fernández-Fígares[2]

[1] Départment Sciences Agronomiques et Animales, Institut Polytechnique LaSalle Beauvais, Beauvais, France
[2] Department of Physiology and Biochemistry of Animal Nutrition, Estación Experimental del Zaidín, Consejo Superior de Investigaciones Científicas, Granada, Spain

Corresponding author
Ignacio Fernández-Fígares,
ifigares@eez.csic.es

## ABSTRACT

As insulin sensitivity may help to explain divergences in growth and body composition between native and modern breeds, metabolic responses to glucose infusion were measured using an intra-arterial glucose tolerance test (IAGTT). Iberian ($n = 4$) and Landrace ($n = 5$) barrows ($47.0 \pm 1.2$ kg body weight (BW)), fitted with a permanent carotid artery catheter were injected with glucose (500 mg/kg BW) and blood samples collected at -10, 0, 5, 10, 15, 20, 25, 30, 45, 60, 90, 120 and 180 min following glucose infusion. Plasma samples were analysed for insulin, glucose, lactate, triglycerides, cholesterol, creatinine, albumin and urea. Insulin sensitivity indices were calculated and analysed. Mean plasma glucose, creatinine and cholesterol concentrations were lower ($P < 0.01$) in Iberian (14, 68 and 22%, respectively) than in Landrace pigs during the IAGTT. However, mean plasma insulin, lactate, triglycerides and urea concentrations were greater ($P < 0.001$) in Iberian (50, 35, 18 and 23%, respectively) than in Landrace pigs. Iberian pigs had larger area under the curve (AUC) of insulin ($P < 0.05$) or tended to a greater AUC of lactate ($P < 0.10$), and a smaller ($P < 0.05$) AUC for glucose 0-60 min compared with Landrace pigs. Indices for estimating insulin sensitivity in fasting conditions indicated improved β-cell function in Iberian compared with Landrace pigs, but no difference ($P > 0.10$) in calculated insulin sensitivity index was found after IAGTT between breeds. A time response ($P < 0.05$) was obtained for insulin, glucose and lactate so that maximum concentration was achieved at 10 and 15 min post-infusion for insulin (Iberian and Landrace pigs, respectively), immediately post-infusion for glucose, and 20 min post-infusion for lactate, decreasing thereafter until basal levels. There was no time effect for the rest of metabolites evaluated. In conclusion, growing Iberian pigs challenged with an IAGTT showed changes in biochemical parameters and insulin response that may indicate an early stage of insulin resistance.

# INTRODUCTION

The Iberian pig is a slow growing native breed of the Mediterranean basin with much greater whole body fat content than lean-type pigs (*Nieto et al., 2002*). Compared with conventional breeds, Iberian pigs show a lower efficiency of energy utilisation for protein
deposition in the growing period (*Barea, Nieto & Aguilera, 2007*). The greater relative viscera weight (*Rivera-Ferre, Aguilera & Nieto, 2005*) and total heat production (*González-Valero et al., 2016*) associated in part with the greater rate of muscle protein turnover (*Rivera-Ferre, Aguilera & Nieto, 2005*) in Iberian compared with lean-type pigs help to explain the low energy efficiency for growth. In fact, *Rivera-Ferre, Aguilera & Nieto (2005)* showed that muscle protein degradation was increased in Iberian pigs resulting in decreased muscle protein accretion compared with Landrace pigs. Interestingly, insulin resistance at the muscle level could explain an increased protein degradation (*Wang et al., 2006*) affecting overall protein accretion. In a previous study using balanced or lysine deficient diets at two crude protein levels, Iberian had greater fasting serum insulin concentration than Landrace pigs (*Fernández-Fígares et al., 2007*), suggesting the possibility of insulin resistance in Iberian pigs. We hypothesised that Iberian pigs have decreased insulin sensitivity, which could explain differences on growth, body composition and metabolic characteristics compared with modern breeds. The objective of the present study was to evaluate differences on insulin sensitivity between Iberian and Landrace pigs using an intra-arterial glucose tolerance test (IAGTT).

## MATERIALS & METHODS

### Animals and experimental design

All procedures used in this study were approved by the Bioethical Committee of the Spanish Council for Scientific Research (RD 53/2013; CSIC, Spain; project reference RECUPERA 2020, FEDER funding) and the animals were cared for in accordance with the Royal Decree No. 1201/2005 (Spain). The experiment was performed with five Landrace and four Iberian (Silvela strain) barrows supplied by Granja El Arenal (Córdoba, Spain) and Sánchez Romero Carvajal (Jabugo S.A., Puerto de Santa María, Cádiz, Spain), respectively.

The pigs were group housed in a controlled-environment room (20 m$^2$; 21 $\pm$ 1.5 °C) with *ad libitum* access to a standard barley-soybean meal diet (160 g crude protein/kg; 14 MJ metabolizable energy/kg dry matter) and water. During acclimatization, the pigs were adapted to close contact with the personnel involved in the study to facilitate pig handling without stress. After acclimatization and to subsequently avoid the stress of repeated blood sampling, each animal was surgically fitted with a chronic catheter (Tygon, i.d. 1.02 mm, o.d. 1.78 mm; Cole-Parmer, Vernon Hills, IL, USA) in the carotid artery following a procedure described previously (*Rodríguez-López et al., 2013*). In brief, the day before surgery, pigs were placed in individual pens (2 m$^2$), where nose and eye contact was possible, in a controlled environment room (21 $\pm$ 1.5 °C) and feed and water removed. General anaesthesia was induced using an intamuscular (i.m.) combination of Ketamine (15 mg/kg BW; Imalgene 1000, Merial, Barcelona, Spain)/Azaperone (2 mg/kg BW; Stresnil, Steve, Barcelona, Spain) and maintained with isoflurane (0.5–2%; Isoflo; Laboratorios Esteve S.A., Barcelona, Spain) and O$_2$ (22-44 mL/kg BW/min) through a face mask. N-butyl hyoscine bromide + Sodium metamizol (Buscapina Compositum; Boehringer Ingelheim Spain S.A., Barcelona, Spain) was administered as analgesic (5 mL i.m.). Strict aseptic and sterile conditions were applied along the whole surgical procedure.

An incision (8–10 cm) was done along the jugular furrow and to expose the carotid artery. Catheter was introduced 12 cm toward the aorta arch and fixed by non-absorbable suture. The catheter was secured directly in place with a purse-string suture where the artery was not occluded. The incision was sutured. A patch ($10\times 10$ cm) was glued to the skin together with the catheter close to the exteriorization point (down and caudal from the ear), guided to the shoulder, fixed again and kept coiled with a second patch-pocket. Following surgery, pigs returned to their individual pens under heat lamps to provide additional warmth during anaesthesia recovery. After that, pigs were fed with free access to water. Feed, water intake and body temperature were monitored during a couple of days. Then, pigs were group housed again until the blood sampling and fed at $2.4\times$ metabolizable energy for maintenance (444 kJ/kg$^{0.75}$ body weight (BW)/day; (*NRC, 1998*). Wounds from surgery and catheter exteriorization site were kept clean and sprayed with antibiotic (Veterin Tenicol; Lab. Intervet S.A., Salamanca, Spain) to prevent infection. Pigs were injected i.m. with a broad spectrum antibiotic (Duphapen Strep; Fort Dodge Vet. S.A., Gerona, Spain) during 5 days (5–10 mg/kg BW/day). After 10 days, stitches were removed and pigs were ready for the blood sampling. Patency of catheter was checked weekly, cleaned with alcohol and flushed with sterile heparinized (Fragmin, 5000 IU/0.2 mL; Pharmacia Spain S.A., Barcelona, Spain) saline (250 IU/mL).

The day before the experiment, pigs were randomly accommodated in the individual pens for easier blood sampling and fed normally. On the day of the experiment, all pigs ($46.0 \pm 3.0$ and $47.8 \pm 3.6$ kg BW for Iberian and Landrace pigs, respectively; that is about 18 and 14 weeks of age, respectively) were given an intra-arterial bolus (500 mg/kg BW) of glucose (50% sterile dextrose; glucosado 50% Braun, B. Braun Medical S.A., Rubi, Barcelona, Spain) over one min period after an overnight fast. The catheter was immediately flushed with five mL of sterile saline solution. Blood samples (five mL) were collected at -10, 0 (20–30 s after the bolus of glucose and the saline solution), 5, 10, 15, 20, 25, 30, 45, 60, 90, 120 and 180 min following glucose infusion. At the end of the study, pigs were slaughtered -in accordance with the Royal Decree No. 1201/2005 (Spain)- by electrical stunning.

The staff involved in the experiment was aware of the group allocation at the different stages of the experiment.

## Biochemical analysis and calculations

Plasma was obtained by centrifugation (4 °C, $1820\times$ g for 30 min; Eppendorf 5810 R, Hamburg, Germany) and stored in aliquots at $-20$ °C until insulin and metabolites (glucose, lactate, triglycerides, cholesterol, creatinine, albumin and urea) were analysed. All samples were assayed in duplicate except for insulin which was assayed in triplicate.

Insulin was measured using commercially-available radioimmuno assay kit following the directions of the manufacturer (Millipore porcine insulin radioimmuno assay kit; Cat. PI-12K). Radioactivity in samples was measured using a gamma counter (Behring 1612; Nuclear Enterprises Ltd, Edinburgh, Scotland). Human insulin was used as standard, and the assay was validated for use in porcine plasma samples (*Fernández-Fígares et al., 2007*). The intra- and inter-assay coefficient of variation for plasma insulin were 4.4 and 9.1%,

respectively. The lowest level of insulin that can be detected by this assay is 1.611 µU/mL when using a 100 µL sample size.

Plasma glucose, lactate, triglycerides, cholesterol, creatinine, albumin and urea were measured colorimetrically using an automated Cobas Integra 400® analyser (Roche Diagnostics GmbH, Mannheim, Germany). Analyses were performed in a single run where intra-assays coefficients of variation were 1.3, 0.92, 1.6, 0.81, 3.1, 1.2 and 2.3% for glucose, lactate, triglycerides, cholesterol, creatinine, albumin and urea, respectively.

Responses of plasma insulin, glucose and lactate were evaluated separately by computing total area under the response curve (AUC) determined using trapezoidal geometry (GraphPad Prism, Version 5.02. San Diego, CA) for the time period indicated following intra-arterial glucose infusion (e.g., AUC0-5 stands for the integrated area between 0-5 min post-infusion, AUC0-10 between 0-10 min post-infusion, and so on, until AUC0-180). Basal levels per breed (at time -10 min) were used to calculate the corresponding AUC per metabolite. The rates of decline in plasma insulin and glucose concentrations for both breeds were calculated based on the slope in the linear portion of the response curve from 0 to 30 min after IAGTT challenge (*Christoffersen et al., 2009*). Results were then expressed as a fractional rate constant determined from the slope of the natural logarithm of plasma concentrations vs. time (*Shipley & Clark, 1972*) (cited by *Gopinath & Etherton, 1989*). The fractional turnover rates ($k$), or disappearance rates, of plasma insulin and glucose (%/min) were calculated using the relationship (*Kaneko, 2008*):

$k = (Ln1 - Ln2)/(T_2 - T_1)$

where Ln1 and Ln2 are the natural logarithms of plasma insulin (µU/mL) or glucose (Mm) concentrations at times $T_1$ (0 min) and $T_2$ (30 min), respectively.

From the $k$ value, the half-life, $T_{1/2}$ (min), may be calculated as:

$T_{1/2} = 100 \times 0.693/k$

For insulin sensitivity, indices used in human medicine were used.

The so-called homeostasis model assessment (HOMA; *Matthews et al., 1985*) was calculated for estimating insulin resistance (HOMA-IR) and β-cell function (HOMA-%B) at fasting conditions, as follows:

HOMA-IR = fasting plasma insulin (µU/mL) × fasting plasma glucose (mM)/22.5

HOMA-%B = (20× fasting plasma insulin (µU/mL))/(fasting plasma glucose (mM) - 3.5)

It is assumed that non-insulin-resistant individuals have 100% β-cell function and an insulin resistance of 1.

The quantitative insulin sensitivity check index (QUICKI; *Katz et al., 2000*) was computed as:

QUICKI = $1/[Ln(I_0) + Ln(G_0)]$

where $I_0$ is the fasting insulin (µU/mL), and $G_0$ is the fasting glucose (mg/dl).

Finally, the insulin sensitivity index (CSI; *Tura et al., 2010*) was calculated as:

CSI = $K_G/(\Delta AUC_{INS}/T)$

where $K_G$ is the slope of Ln glucose in the linear portion of the response curve, $\Delta AUC_{INS}$ is the AUC of insulin above basal value, and T is the time interval (between 0 and 30 min) when $K_G$ and $\Delta AUC_{INS}$ are calculated.

## Statistical analyses

The number of animals was *a priori* calculated using the G*Power software (Heinrich-Heine-Universität Düsseldorf (*Faul et al., 2007*)). Accepting an alpha risk of 0.05 and a beta risk of 0.2 in a two-sided test, five subjects are necessary in first group and five in the second to recognize as statistically significant a difference greater than or equal to 12 μU/mL on insulin concentration and a common standard deviation of 6.3 μU/mL based on previous studies (*Fernández-Fígares et al., 2007*). A total of five pigs per treatment was also used by others (e.g., *Stoll et al., 1999*). However, one Iberian pig lost the arterial catheter during the recovery period after surgery and only four Iberian pigs could be used.

Plasma metabolites were evaluated using a mixed ANOVA with repeated measures (Version 9.4; PROC MIXED, SAS Institute Inc., Cary, NC, USA) with the fixed effects of breed, time of sampling and their interaction in the model statement. The pig was considered the experimental unit and a random effect. First-order ante dependence covariance ANTE(1) was used, which allows unequal variances over time and unequal correlations and covariance among different pairs of measurements. Plasma concentration differences between breeds at each sampling time were analysed by the pdiff (piecewise differentiable) option.

Assumptions that are required for an ANOVA were tested following the protocol from *Zuur, Ieno & Elphick (2010)*. Homogeneity of variance was assured by applying the Levene's-Test. No transformation was required. Least square means and pooled standard error of mean (SEM) are presented. Outliers were identified and removed when the absolute studentized residues exceeded 3. Differences were considered significant at $P < 0.05$ and trends approaching significance were considered for $0.05 < P < 0.10$.

## RESULTS

Average plasma metabolites and insulin concentrations after the IAGTT are shown in Table 1. Mean plasma glucose, cholesterol and creatinine concentrations were lower in Iberian (14, 22 and 68%, respectively; $P < 0.05$) compared with Landrace pigs. However, mean plasma insulin, lactate, triglycerides and urea concentrations were greater in Iberian (50, 35, 18 and 23%, respectively; $0.01 < P < 0.001$) than in Landrace pigs. No differences ($P > 0.10$) were found between breeds for albumin levels.

Fasting plasma insulin was greater in Iberian compared with Landrace pigs ($P < 0.05$; Table 2) whereas fasting plasma glucose was similar for both breeds ($P > 0.10$; Table 2). No differences between breeds were found in fasting plasma albumin (Iberian 0.50 and Landrace 0.54 μM), urea (Iberian 3.3 and Landrace 3.0 mM), cholesterol (Iberian 1.46 and Landrace 1.79 mM) and triglycerides (Iberian 0.28 and Landrace 0.22 mM). On the other hand plasma fasting creatinine was lower in Iberian pigs compared to Landrace (54 and 102 μM, SEM = 8.18, respectively; $P < 0.01$).

Only plasma insulin (Fig. 1), glucose (Fig. 2) and lactate (Fig. 3) concentrations changed throughout time ($P < 0.001$; Table 1) after the IAGTT.

An interaction between breed and time was found for plasma insulin, such that concentration of insulin was greater in Iberian pigs from -10–15 min and from 90–180 min

**Table 1** Average plasma metabolites and insulin concentrations in Iberian ($n = 4$) and Landrace ($n = 5$) pigs during an intra-arterial glucose challenge (IAGTT; 500 mg/kg BW, 0–180 min)[a].

| | Breed | | | P-value[b] | | |
|---|---|---|---|---|---|---|
| | Iberian | Landrace | SEM[c] | Breed | Time | Breed x Time |
| Insulin (µU/mL) | 41 | 27 | 1.9 | *** | *** | *** |
| Glucose (mmol/L) | 6.8 | 7.7 | 0.26 | ** | *** | ns |
| Lactate (mmol/L) | 1.3 | 1.0 | 0.039 | *** | *** | ns |
| Triglycerides (mmol/L) | 0.28 | 0.24 | 0.009 | ** | ns | ns |
| Cholesterol (mmol/L) | 1.5 | 1.8 | 0.033 | *** | ns | ns |
| Creatinine (µmol/L) | 54 | 90 | 1.2 | ** | ns | ns |
| Albumin (mmol/L) | 0.48 | 0.50 | 0.009 | ns | ns | ns |
| Urea (mmol/L) | 3.0 | 2.4 | 0.102 | *** | ns | ns |

**Notes.**
[a] Average ($n = 9$) basal level of each metabolite: Glucose = 5.33 mM, Insulin = 11.43 µU/mL, Lactate = 0.87 mM, Triglycerides = 0.25 mM, Albumin = 0.46 mM, Cholesterol = 1.63 mM, Creatinine = 76.2 µM, Urea = 2.92 mM.
[b] ns = non-significant.
[c] Standard error of mean.
** $P < 0.01$.
*** $P < 0.001$.

**Table 2** Indices of glucose tolerance and insulin sensitivity in Iberian ($n = 4$) and Landrace ($n = 5$) pigs subjected to an intra-arterial glucose tolerance test[a].

| | Breed | | | |
|---|---|---|---|---|
| | Iberian | Landrace | SEM[b] | P-value[c] |
| Fasting insulin (µU/mL) | 16 | 8 | 1.6 | * |
| Fasting glucose (mmol/L) | 4.7 | 5.9 | 0.92 | ns |
| Insulin disappearance rate (%/min) | 7.2 | 3.8 | 1.01 | † |
| Glucose disappearance rate (%/min) | 5.9 | 3.9 | 0.77 | † |
| Insulin half-live (min) | 10 | 21 | 3.6 | † |
| Glucose half-live (min) | 22 | 12 | 4.8 | ns |
| QUICKI | 0.31 | 0.33 | 0.007 | * |
| HOMA-IR | 3.3 | 2.3 | 0.58 | ns |
| HOMA-%B | 267 | 100 | 26 | ** |
| CSI ($\times 10^{-4}$) | −12 | −13 | 1.8 | ns |

**Notes.**
[a] QUICKI, quantitative insulin sensitivity check index; HOMA-IR, homeostasis model assessment for estimating insulin resistance; HOMA-%B, homeostasis model assessment for estimating -cell function; CSI, calculated insulin sensitivity index.
[b] Standard error of mean.
[c] ns = non-significant.
† $0.05 < P < 0.10$.
* $P < 0.05$.
** $P < 0.01$.

($P < 0.05$, with $P < 0.10$ at times 0 and 90 min) and lower at 25 min ($P < 0.10$; Fig. 1). In both breeds, plasma insulin levels increased 7-fold, reaching a peak concentration at 10 and 15 min after glucose infusion for Iberian ($113.6 \pm 7.1$ µU/mL) and Landrace ($55.7 \pm 6.4$ µU/mL) pigs, respectively. Insulin remained well above fasting levels until 20 and 45 min after glucose infusion for Iberian and Landrace pigs, respectively; thereafter insulin levels rapidly decreased until fasting levels were attained. Insulin disappearance rate tended to

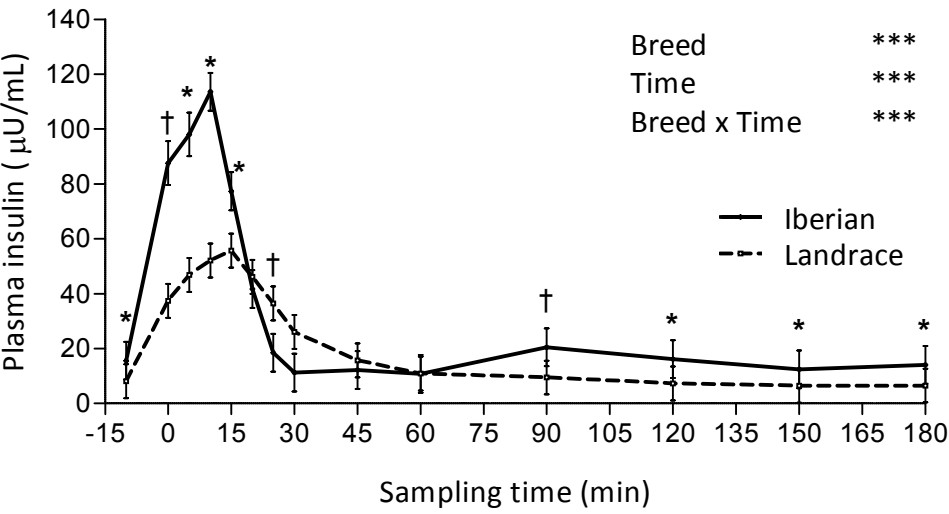

**Figure 1** Plasma insulin concentration during intra-arterial glucose challenge test (500 mg/kg BW; 180 min sampling) in growing Iberian ($n = 4$) and Landrace ($n = 5$) pigs. †$0.05 < P < 0.10$, * $P < 0.05$, *** $P < 0.001$.

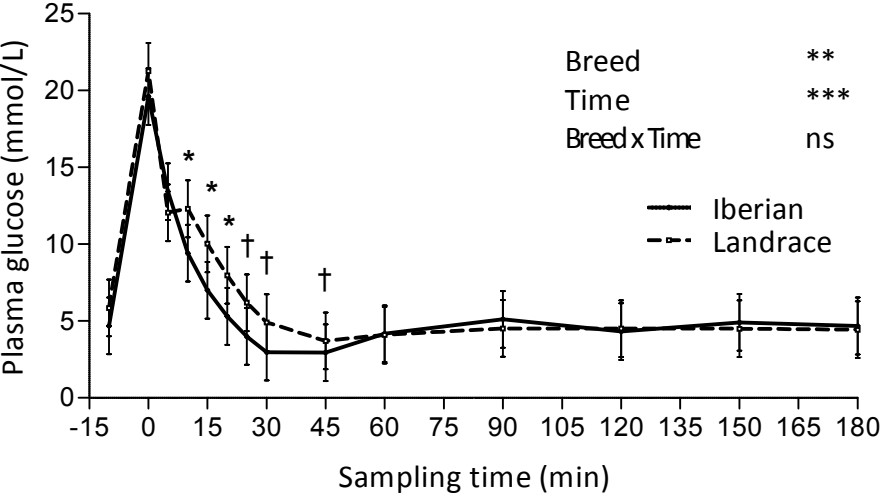

**Figure 2** Plasma glucose concentration during intra-arterial glucose challenge test (500 mg/kg BW; 180 min sampling) in growing Iberian ($n = 4$) and Landrace ($n = 5$) pigs. ns, not significant ($P > 0.10$); †$0.05 < P < 0.10$, * $P < 0.05$, ** $P < 0.01$, *** $P < 0.001$.

increase in Iberian compared with Landrace pigs ($0.05 < P < 0.10$) while insulin half-life tended to decrease ($0.05 < P < 0.10$; Table 2).

Glucose peaked (Fig. 2) immediately after glucose infusion reaching a value of 19.6 and 21.2 mmol/L for Iberian and Landrace pigs, respectively. Subsequently, glucose concentration gradually decreased to values below fasting levels after 25 and 30 min, respectively for Iberian and Landrace pigs. The lowest plasma glucose concentration (glucose nadir) was found at 45 min (2.95 and 3.70 mmol/L for Iberian and Landrace pigs,

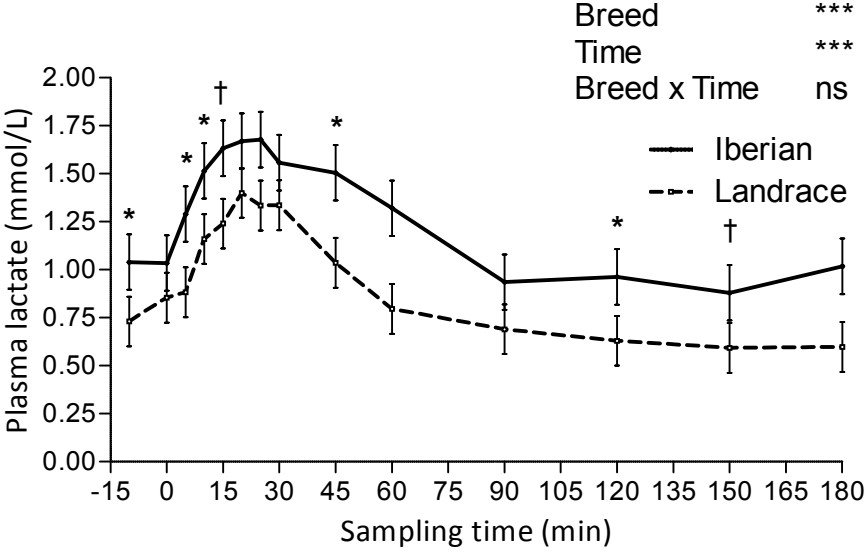

**Figure 3** Plasma lactate concentration during intra-arterial glucose tolerance test (500 mg/kg BW; 180 min sampling) in growing Iberian ($n = 4$) and Landrace ($n = 5$) pigs. ns, not significant ($P > 0.10$); †$0.05 < P < 0.10$, * $P < 0.05$, *** $P < 0.001$.

respectively). After glucose nadir, glucose concentration gradually increased again to reach values comparable to fasting levels at 180 min. No differences were found between breeds for glucose disappearance rate ($P > 0.10$; Table 2) or glucose half-life ($P > 0.10$; Table 2).

Lactate increased after the IAGTT, peaked at 20 min for both breeds and declined progressively until reaching basal concentrations at 180 min (Fig. 3).

The AUC values for each sampling time of insulin, glucose and lactate are shown in Tables 3, 4 and 5, respectively. Insulin AUC was greater ($P < 0.05$) for Iberian compared with Landrace pigs at all times.

Conversely, glucose AUC at 0–15, 0–20, 0–25, 0–30, 0–45 and 0–60 min were lower ($P < 0.05$) for Iberian than Landrace pigs. Aditionally, glucose AUC tended to be lower ($0.05 < P < 0.10$) at 0–10, 0–90 and 0–120 min. Plasma lactate AUC was greater ($P < 0.05$) for Iberian pigs at 0–10 and 0–15 min and tended to be greater ($0.05 < P < 0.10$) at 0–5, 0–20, 0–25, 0–90, 0–120, 0–150 and 0–180 min after glucose infusion.

Indices of insulin sensitivity are shown in Table 2. The QUICKI index decreased ($P < 0.05$) while HOMA-%B index increased ($P < 0.01$) in Iberian compared with Landrace pigs. No differences ($P > 0.10$) were found for HOMA-IR and CSI.

## DISCUSSION

The IAGTT method allowed the comparison of the insulin responsiveness of an obese (Iberian) and a lean (Landrace) pig breed. It is well established that Iberian pigs have a much greater capacity of lipid deposition in comparison to lean swine breeds (*Nieto et al., 2002*; *Ovilo et al., 2005*; *Muñoz et al., 2009*) and has since been proposed as a pig model for obesity studies (*Rodriguez Rodriguez et al., 2020*). For the reason that important differences

**Table 3** Area under the curve (AUC, U ×min/mL) of plasma insulin during intra-arterial glucose tolerance test between minute 0 and indicated time post-challenge in Iberian ($n = 4$) and Landrace ($n = 5$) pigs.

|  | Iberian | Landrace | SEM[a] | P- value[b] |
|---|---|---|---|---|
| AUC 0-5 min | 464 | 295 | 40.7 | * |
| AUC 0-10 min | 993 | 587 | 72.6 | ** |
| AUC 0-15 min | 1470 | 873 | 93.3 | ** |
| AUC 0-20 min | 1768 | 1135 | 95.1 | ** |
| AUC 0-25 min | 1919 | 1335 | 92.5 | ** |
| AUC 0-30 min | 1993 | 1461 | 89.0 | ** |
| AUC 0-45 min | 2169 | 1652 | 106.3 | ** |
| AUC 0-60 min | 2340 | 1757 | 135.1 | * |
| AUC 0-90 min | 2809 | 1985 | 176.3 | * |
| AUC 0-120 min | 3358 | 2239 | 206.5 | ** |
| AUC 0-150 min | 3787 | 2459 | 214.3 | ** |
| AUC 0-180 min | 4183 | 2642 | 229.1 | ** |

Notes.
[a] Standard error of mean.
[b] $P < 0.05$.
** $P < 0.01$.

**Table 4** Area under the curve (AUC, mmol ×min/L) of plasma glucose during intra-arterial glucose tolerance test between minute 0 and indicated time post-challenge in Iberian ($n = 4$) and Landrace ($n = 5$) pigs.

|  | Iberian | Landrace | SEM[a] | P- value[b] |
|---|---|---|---|---|
| AUC 0–5 min | 82.6 | 90.8 | 3.94 | ns |
| AUC 0–10 min | 140 | 159 | 7.6 | † |
| AUC 0–15 min | 181 | 215 | 10.4 | * |
| AUC 0–20 min | 211 | 260 | 12.9 | * |
| AUC 0–25 min | 235 | 295 | 15.7 | * |
| AUC 0–30 min | 252 | 323 | 18.7 | * |
| AUC 0–45 min | 297 | 388 | 25.7 | * |
| AUC 0–60 min | 350 | 446 | 28.3 | * |
| AUC 0–90 min | 489 | 575 | 30.4 | † |
| AUC 0–120 min | 631 | 710 | 30.0 | † |
| AUC 0–150 min | 769 | 845 | 30.2 | ns |
| AUC 0–180 min | 913 | 979 | 31.7 | ns |

Notes.
[a] Standard error of mean.
[b] ns = non-significant.
† $0.05 < P < 0.10$.
* $P < 0.05$.

in for instance protein turnover may take place during development (*Lobley, 1993*), the study of animal breeds with disparate growth capability is not a simple issue. Thus, it is desirable that animals are comparable for age or physiological state. As the developmental age of the animals may vary, a decision was made regarding the use pigs of the same BW considering that age difference at this early state was acceptable.

**Table 5 Area under the curve (AUC, mmol × min/L) of plasma lactate during intra-arterial glucose tolerance test between minute 0 and indicated time post-challenge in Iberian ($n = 4$) and Landrace ($n = 5$) pigs.**

|  | Iberian | Landrace | SEM[a] | P-value[b] |
|---|---|---|---|---|
| AUC 0–5 min | 5.81 | 4.34 | 0.473 | † |
| AUC 0–10 min | 12.8 | 9.4 | 0.91 | * |
| AUC 0–15 min | 20.7 | 15.5 | 1.45 | * |
| AUC 0–20 min | 29.0 | 22.1 | 2.23 | † |
| AUC 0–25 min | 37.3 | 28.9 | 3.09 | † |
| AUC 0–30 min | 45.4 | 35.6 | 4.00 | ns |
| AUC 0–45 min | 68.4 | 53.3 | 6.87 | ns |
| AUC 0–60 min | 89.6 | 67.1 | 9.29 | ns |
| AUC 0–90 min | 123 | 89.4 | 13.01 | † |
| AUC 0–120 min | 152 | 109 | 15.7 | † |
| AUC 0–150 min | 180 | 128 | 17.6 | † |
| AUC 0–180 min | 208 | 145 | 20.7 | † |

Notes.
[a] Standard error of mean.
[b] ns = non-significant.
† $0.05 < P < 0.10$.
* $P < 0.05$.

As arterial blood represents the metabolites concentration to which the tissues are exposed (*Brouns et al., 2005*), chronic catheters were inserted in carotid artery for glucose infusion and blood sampling.

In the current study we have shown that despite the higher fasting plasma insulin, Iberian pigs produce a higher insulin response after glucose infusion when compared to Landrace pigs (18 and 14 weeks of age, respectively). Greater postprandial serum levels of insulin have been described in 20 kg BW Iberian (11 weeks of age) compared to Landrace pigs after glucose infusion (*Fernández-Fígares et al., 2007*), and in 11 kg BW Ossabaw (obese; 10 weeks of age) compared to 16.5 kg Yorkshire (10 weeks of age) pigs (*Wangsness et al., 1981*). However, other comparative studies using a standard diet found increased insulin secretion in 75-120 kg BW Large White boars than in 40-75 kg BW Meishan boars (obese breed) at 20 and 52 weeks of age, respectively (*Weiler et al., 1998*). The limited growth and development of slow growing pigs could result at least partly from disturbances in insulin secretion and/or in insulin binding, leading to insulin sensitivity, because most cells of the body require insulin for adequate uptake of glucose and amino acids (*Claus & Weiler, 1994*). If the concentration of insulin is compared among animals of different breeds, the sensitivity of each breed to insulin should be considered. In this study we were also interested in other key metabolites which could provide additional information concerning reduced insulin sensitivity in Iberian pigs. After glucose infusion, glucose plasma concentration rapidly returned to preprandial values in the present experiment, which indicates that exogenous glucose was efficiently metabolized, stored as glycogen, or both. As expected, when glucose was infused, plasma glucose levels were rapidly increased and a subsequent insulin response was observed. The elevated insulin lowered plasma glucose below fasting values within 20 and 25 min for Iberian and Landrace pigs, respectively, and insulin levels returned to

baseline as plasma glucose declined. In our study, glucose concentration and glucose AUC during the IAGTT were lower in Iberian compared with Landrace pigs, with no differences in fasting plasma glucose, maybe due to the limited number of pigs. When interpreting the individual glucose curves, a monophasic pattern was identified for both breeds. The lower glucose AUC of Iberian pigs ($-19\%$ on average) may be related to the greater insulin AUC ($+33\%$ on average), a common pattern in many models of obesity (*Kay et al., 2001*). However, the reasons for the unequal physiological response between breeds are not well understood and must be discussed.

As it has been proved that the energy needs of portal-drained viscera are fulfilled by the oxidation of glucose, glutamate, and glutamine in pigs (*Stoll et al., 1999*), a larger gastrointestinal tract of Iberian pigs compared to Landrace (*Rivera-Ferre, Aguilera & Nieto, 2005*) is in line with the decreased AUC of glucose reported in our experiment.

However, despite the larger size of the gastrointestinal tract and lower portal blood flow (*González-Valero et al., 2016*) of Iberian compared with Landrace pigs, no differences on net portal flux of glucose after ingestion of the same diet were found (*Rodríguez-López et al., 2013*). Differences on insulin stimulated glucose transport at portal-drained viscera level may help to explain these results. Iberian have lower glucose concentrations than Landrace pigs after an intravenous adrenaline challenge (*Fernández-Fígares et al., 2016*), suggesting a decreased response of Iberian pigs to sympathetic nervous system stimuli which is in line with the lower glucose AUC reported here.

When insulin sensitivity indices used in human medicine were applied to the conditions of the present experiment, QUICKI and HOMA-%B were more sensitive detecting differences between breeds. Indeed, QUICKI index decreased in Iberian compared with Landrace pigs, pointing out an incipient insulin sensitivity impairment in fasting Iberian pigs. Similarly, reduced QUICKI index (0.5 vs. 0.6) was found in Bama miniature pigs fed a high sucrose and fat diet compared with a control diet, respectively (*Liu et al., 2017*). The QUICKI index has been shown to provide reasonable approximations of insulin efficiency in minipigs (*Christoffersen et al., 2009*).

When we used the HOMA, differences on hepatic HOMA-IR were negligible between breeds (3.3 and 2.3 for Iberian and Landrace pigs, respectively; $P > 0.10$). However, Iberian had improved β-cell function compared with Landrace pigs according to HOMA-%B (267 and 100, respectively; $P < 0.01$), which may be due to enhanced sensitivity of the β-cells to glucose during the fasting period. As a consequence, β-cell insulin synthesis in Iberian pigs increased in accordance with the increased insulin release after the glucose tolerance test and the elevated basal insulin concentrations reported for Iberian pigs. This is consistent with decreased QUICKI in Iberian pigs compared to Landrace (0.31 and 0.33, respectively; $P < 0.05$).

Lactate appearance after an intravenous glucose test is positively associated with insulin sensitivity in humans (*Lovejoy et al., 1992*), as it is related to lactate production by insulin sensitive tissues (mainly muscle and fat). Because only limited amounts of lactate are produced by muscle after glucose loading (*Ykijarvinen, Bogardus & Foley, 1990*), the source of lactate appearance should predominantly be adipose tissue (*Lovejoy et al., 1992*), with a large capacity to convert glucose to lactate (*Marin et al., 1987*). We report here a delay of

20 min in plasma lactate elevation relative to glucose peak following IAGTT, which may reflect the time lag in adipose tissue uptake of glucose and subsequent lactate production under the stimulation of insulin. Compared with Landrace, the increased lactate AUC in Iberian pigs after the IAGTT could therefore be a consequence of the greater adipose tissue availability (*Nieto et al., 2002*) instead of greater insulin sensitivity. On the other hand, insulin resistance was associated with elevated basal lactate levels in obese humans (*Lovejoy, Mellen & Digirolamo, 1990*), so increased basal lactate concentrations in Iberian pigs (1.040 vs. 0.730 mmol/L; SEM = 0.063) could also indicate insulin resistance or reduced insulin sensitivity. Although inhibition of insulin action on glycogenolysis in fasting conditions may lead to increased glucose release from glycogen and subsequent conversion of glucose to lactate, there is no direct evidence of this. There is indirect evidence, though, that elevated lactate levels could be a consequence of greater adipose tissue availability and may also reflect a glucose sparing effect (decreased glucose utilisation) in muscle (*Pearce & Connett, 1980*).

Obesity is frequently associated with different degrees of dyslipidemia manifested as increased triglyceridemia and low HDL-cholesterol. In our experiment, we found lower plasma total cholesterol but greater plasma triglycerides concentration in Iberian compared with Landrace pigs. Although we did not separate LDL and HDL fractions, total cholesterol concentration are phenotypically related with LDL and HDL cholesterol concentrations in pigs (*Rauw et al., 2007*). Reduced total cholesterol concentration could be due to reduced hepatic insulin sensitivity as insulin stimulates cholesterol synthesis (*Nelson & Cox, 2017*). In any case the cholesterolemia for both breeds in the present experiment was in the lower range of published values (*Fernández-Fígares et al., 2007*). Indeed, the pigs in this study were still very young and so a greater level should be expected at a later stage of development (*Rauw et al., 2007*).

Previous studies in our lab have shown the low genetic potential of growing Iberian pigs for muscle protein deposition in comparison to lean breeds (*Nieto et al., 2002*), possibly due to the greater muscle protein degradation and turnover of the former (*Rivera-Ferre, Aguilera & Nieto, 2005*). In line with this, plasma urea level (an indirect protein degradation indicator) was in the present study 23% greater in Iberian compared with Landrace pigs. Differences on circulating insulin or the capacity of insulin release between breeds may explain differences in lean tissue deposition, as insulin has an important role in skeletal muscle metabolism (*Wang et al., 2006*). In obese db/db mice (a model of insulin deficiency) higher muscle protein degradation in comparison with control mice (normal plasma insulin concentration) was reported; the authors concluded that insulin resistance was associated with accelerated muscle protein degradation (*Wang et al., 2006*). The elevated protein degradation reported in Iberian compared with Landrace pigs (*Rivera-Ferre, Aguilera & Nieto, 2005*) suggests the possibility of insulin resistance at this level. The lower plasma creatinine level (indicator of muscle mass) found in this study for Iberian pigs is in accordance with previous studies (*Fernández-Fígares et al., 2007*) and also with the low muscle protein deposition and muscle size described previously (*Nieto et al., 2002*; *Rivera-Ferre, Aguilera & Nieto, 2005*). As insulin resistance is associated with decreased muscle mass, plasma creatinine levels can also be used as an indicator of insulin

signalling disorders as reported by *Kashima et al. (2017)* in humans. Further research regarding amino acids concentration after an IAGTT may help to explain differences in the effect of insulin on muscle protein metabolism between breeds.

Previous studies from our lab indicate that growing Iberian pigs are prone to insulin resistance compared with modern breeds as denoted by increased hepatic gluconeogenesis (*González-Valero et al., 2014*), greater plasma free fatty acid concentration (*Fernández-Fígares et al., 2016*) and lower plasma creatinine and QUICKI (*Fernández-Fígares et al., 2007*). Additionally, in this experiment we show greater HOMA-%B and increased plasma insulin and lactate concentrations after an IAGTT. The increased plasma insulin AUC after an IAGTT suggests insulin resistance in comparison to the values obtained for lean pigs, although the concentration of glucose remained low which could indicate the absence of a peripheral insulin resistance. Although Iberian pigs may be considered an obese breed in terms of body composition (*Nieto et al., 2002*; *Barea, Nieto & Aguilera, 2007*), insulin resistance mechanisms have not yet been fully established at the development stage of the pigs in this experiment. Insulin resistance and impaired glucose tolerance has been shown in Iberian sows (2.5 years old) *ad libitum* fed a saturated fat enriched diet for three months (*Torres-Rovira et al., 2012*).

## CONCLUSIONS

Although our results support the existence of an insulin resistance or a decreased insulin sensitivity in growing Iberian pigs, caution should be taken because of the reduced number of pigs used. The utilization of the hyperinsulinemic euglycemic clamp, the most definitive approach to determine whole-body insulin action should provide conclusive evidence regarding the establishment of insulin resistance in growing Iberian pigs.

## ACKNOWLEDGEMENTS

The authors thank the company Sánchez Romero Carvajal (Jabugo S.A., Puerto de Santa María, Spain) for their helpful collaboration, Dr Luis Lara for statistical advice and Dr Thomas J. Caperna for critically reading the manuscript. A preprint version of this manuscript has been peer-reviewed and recommended by Peer Community In Animal Science (https://doi.org/10.24072/pci.animsci.100004).

### Funding

This research was supported by grants no. AGL 2006-05937/GAN and AGL 2009-08916 from Ministerio de Educación y Ciencia (Spain) and RECUPERA 2020, FEDER funding). There was no additional external funding received for this study. The publication fee was supported by the CSIC Open Access Publication Support Initiative through its Unit of Information Resources for Research (URICI). The funders had no role in study design, data collection and analysis, decision to publish, or preparation of the manuscript.

## Grant Disclosures

The following grant information was disclosed by the authors:

Ministerio de Educación y Ciencia (Spain) and RECUPERA 2020, FEDER: AGL 2006-05937/GAN, AGL 2009-08916.

## Competing Interests

The authors declare there are no competing interests.

## Author Contributions

- José Miguel Rodríguez-López performed the experiments, analyzed the data, prepared figures and/or tables, authored or reviewed drafts of the paper, and approved the final draft.
- Manuel Lachica and Ignacio Fernández-Fígares conceived and designed the experiments, performed the experiments, analyzed the data, prepared figures and/or tables, authored or reviewed drafts of the paper, and approved the final draft.
- Lucrecia González-Valero performed the experiments, analyzed the data, authored or reviewed drafts of the paper, and approved the final draft.

## Animal Ethics

The following information was supplied relating to ethical approvals (i.e., approving body and any reference numbers):

All procedures used in this study were approved by the Bioethical Committee of the Spanish Council for Scientific Research (CSIC, Spain) and the animals were cared for in accordance with the Royal Decree No. 1201/2005 (Spain).

## Ethics

The following information was supplied relating to ethical approvals (i.e., approving body and any reference numbers):

The Bioethical Committee of the Spanish National Research Council (CSIC, Spain) granted Ethical approval to carry out the study within its facilities (project reference RECUPERA 2020).

## Data Availability

Raw data is available as a Supplemental File.

## Supplemental Information

Supplemental information for this article can be found online at http://dx.doi.org/10.7717/peerj.11014#supplemental-information.

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
