# Peer review of "Determining insulin sensitivity from glucose tolerance tests in Iberian and landrace pigs"

_PeerJ, doi:10.7717/peerj.11014_

## Round 0.1 · original submission · Major Revisions

Please take into consideration the reviewer’s comments and provide back a point-by-point rebuttal letter addressing those concerns.

Please be encouraged by the prompt reviews, and we look forward to your revised manuscript.

Reviewer 1 ·

Basic reporting

The manuscript is well written and with clear language despite minor english issues occurring throughtout the paper. Some examples that can improve its simplicity and readability are listed bellow:

line 10 - add "at" before 10 (also happens in line 198).
lines 42-43 - remove "compared with modern lean breeds".
line 61-64 - rewrite as follows: "After acclimatization and to subsequently avoid the stress of repeated blood sampling, each animal was surgically fitted with a chronic catheter (Tygon, i.d. 1.02 mm, o.d. 1.78 mm; Cole-Parmer, Vernon Hills, IL, USA) in the carotid artery following a procedure described previously (Rodríguez-López et al., 2013). "
line 68 - write the abreviation in full extense the first time it is used (also happens with the coefficent of variation, CV)
line 75 - rewrite to: (...) "and to expose the carotid artery."
lines 238-240 - This sentence doesn’t make sense as it is since fasting plasma insulin levels cannot be affected by something that didnt occur yet (IAGTT). I would suggest rewriting as follows: “In the current study we have shown that despite the higher fasting plasma insulin, Iberian pigs produce a higher insulin response after glucose infusion when compared to Landrace pigs.”
line 248 - i assume a "reduced" is missing before insulin sensitivity.
lines 289-290 - i would suggest changing to "be a consequence of the greater adipose tissue availability"
line 350 - add "the" before "values obtained for lean pigs.

The references used are somewhat old but fulfill the requirements for context in the topics discussed. All submitted tables and figures are well constructed and usefull but lack a title. I would also personally suggest for the fasting values to be added to table 1 so that comparing the effect of the before and after is easier on the reader perspective. I would also suggest "pigs" to be used in the title of the manuscript instead of "pig".

Experimental design

The aim of the study is to add new information about the previously suggested idea that insulin resistance or lower insulin sensitivity is occuring in Iberian pigs when compared with leaner breeds. The research methods used fulfill the basic requirements but as stated by the authors in the conclusions, the usage of an hyperinsulinemic euglycemic clamp would provide further knowledge regarding this topic. On the other hand, the authors also state that they didn't separate the LDL and HDL fractions which would help to have a better understanding to what is happening to the total cholesterol concentration. Furthermore isnt it also possible that the lower cholestol levels found in both breeds is related to the fact that they are still very young and could manifest the expected cholesterol changes at a later stage of their development? i think this aspect could have been more discussed...

Validity of the findings

The replication count is barely minimal to extract big conclusions and one of the breeds has one less replicate which i think is a good policy from the authors to explain how this occurred in the manuscript. The data provided are robust as they stand and the conclusions in line to what the results show.

Additional comments

no comment

Reviewer 2 ·

Basic reporting

The current manuscript provides information on insulin sensitivity in two breeds of pigs, Iberian (a slow growing native breed of the Mediterranean basin) and Landrace (conventional) growing pigs. The authors hypothesized that Iberian pigs may be insulin resistant. The present paper is clearly written. Nevertheless, some methodological issues must be addressed and the result and discussion sections need to be improved significantly.

The current study is based on the investigation of 4 or 5 pigs/ breed which is quite low. In this context, we need more information on pig features. The authors should provide at least some data related to body composition (backfat thickness at least to assess adiposity). As the authors refer very often to obesity models in the discussion, we need to have more information on the 9 examined pigs.

Another major concern is dealing with result presentation and also with the analysis of the response to the glucose tolerance test. Data related to fasting plasma concentrations should be included in a table (table 2 or a new table). Similarly, other parameters (L199-201) should be also included in a table. It will facilitate the interpretation of data. There is redundant information on figures. Instead of showing Fig 4-6, it would be more relevant to determine the total AUC and to include it in a table. It should also shorten the result section.

With respect to the discussion, some parts are written like a review of the literature. For instance, it is unclear whether the authors refer to previous experiments or to their study (L247-250). The authors suggest the possibility of insulin resistance in Iberian pigs. To clarify the discussion, they should discuss together all the data assessing insulin sensitivity. There is also an issue with the age of pigs. This point needs to be considered in the discussion.

Other specific comments

Abstract
- L12: we should read “than in Landrace…
- L15: lactate only tended to differ between the two breeds. This point needs to be corrected.

Materials and Methods
- L113: the sensitivity (minimum amount or concentration detectable) of the assay should be provided.
- L114-116: it is necessary to give information on intra- and inter-assay variability.

Discussion
- L227: what are the criteria used to define the obesity of Iberian pigs.

Experimental design

As indicated above, more information on the phenotype of animals is requested.

Validity of the findings

The findings are valid.

Additional comments

The current manuscript is clearly written but needs to be significantly improved.

External reviews were received for this submission. These reviews were used by the Editor when they made their decision, and can be downloaded below.

---

## Round 0.2 · Minor Revisions

Please take into consideration the reviewer’s comments and provide back a point-by-point rebuttal letter addressing those concerns.

Reviewer 2 ·

Basic reporting

No comment

Experimental design

No comment

Validity of the findings

No comment

Additional comments

In the current revised manuscript, the authors incorporated the requested changes with one exception. As indicated previously, there was redundant information on figures. Instead of showing Fig 4-6, I indicated that it would be more relevant to determine the total AUC and to include it in a table. It should also shorten the result section. The authors replaced the 3 figures by 3 tables but did not calculate the total AUC to get one value/animal. This point should be considered.

External reviews were received for this submission. These reviews were used by the Editor when they made their decision, and can be downloaded below.

---

## Round 0.3 · accepted · Accept

Thanks for addressing all the revisions and corrections requested. Now your manuscript is accepted in PeerJ.

External reviews were received for this submission. These reviews were used by the Editor when they made their decision, and can be downloaded below.